# Voluntary wheel running has no impact on brain and liver mitochondrial DNA copy number or mutation measures in the PolG mouse model of aging

Kendra D. Maclaine[1], Kevin A. Stebbings[1,2], Daniel A. Llano[1,2,3], Justin S. Rhodes[1,2,4]*

1 Beckman Institute, University of Illinois at Urbana-Champaign, Urbana, Illinois, United States of America, 2 Neuroscience Program, Beckman Institute, University of Illinois at Urbana-Champaign, Urbana, Illinois, United States of America, 3 Department of Molecular and Integrative Physiology, University of Illinois at Urbana-Champaign, Urbana, Illinois, United States of America, 4 Department of Psychology, University of Illinois at Urbana-Champaign, Champaign, Illinois, United States of America

☯ These authors contributed equally to this work.
* jrhodes@illinois.edu

**Data Availability Statement:** All relevant data are within the paper and its Supporting Information files.

## Abstract

The mitochondrial theory of aging attributes much of the aging process to mitochondrial DNA damage. The polymerase gamma (PolG) mutant mouse was designed to evaluate this theory and thus carries a mutated proofreading region of polymerase gamma (D257A) that exclusively transcribes the mitochondrial genome. As a result, PolG[D257A] mice accumulate mitochondrial DNA (mtDNA) mutations that lead to premature aging, as evidenced by hair loss, weight loss, kyphosis, increased rates of apoptosis, organ damage, and an early death, occurring around 12 months of age. Research has shown that exercise decreases skeletal muscle mtDNA mutations and normalizes protein levels in PolG mice. However, brain mtDNA changes with exercise in PolG mice have not been studied. We found no effects of exercise on mtDNA mutations or copy number in either the brain or liver of PolG mice, despite changes to body mass. Our results suggest that mitochondrial mutations play little role in exercise-brain interactions in the PolG model of accelerated aging. In addition to evaluating the effect of exercise on mtDNA outcomes, we also implemented novel methods for both extracting mtDNA and measuring mtDNA mutations, with aims for improving the efficiency and accuracy of these methods.

## 1. Introduction

The mitochondrial theory of aging attributes much of the aging process to mitochondrial DNA damage. The polymerase gamma D257A (PolgA[D257A/D257A]) mutant mouse, hereafter referred to simply as "PolG", was designed to evaluate the mitochondrial theory of aging and carries a mutated proofreading region of polymerase gamma that exclusively replicates the mitochondrial genome [1]. As a result, PolG mice accumulate mitochondrial DNA (mtDNA)

**Funding:** This project was funded by grant AG059103 (to D.A.L.) and MCB SROP, Jenner award and Beckman Neurotechnology for Memory and Cognition undergraduate fellowship (to K.D. M.). The funders had no role in study design, data collection and analysis, decision to publish, or preparation of the manuscript.

**Competing interests:** The authors have declared that no competing interests exist.

mutations that lead to premature aging phenotypes including hair loss, weight loss, kyphosis, increased rates of apoptosis, organ damage, and early death, occurring around 12 months of age [2–4]. It has been reported that PolG animals have a depletion of mtDNA copy number [4], or the number of mtDNA molecules per nuclear genome, which is generally associated with a decrease in mitochondria, and subsequently, energy production [4,5].

Along with its many other benefits, cardiovascular exercise is one of the only interventions which has consistently been shown to mitigate declines in age-related cognitive function [6–8]. However, the mechanisms behind the impact of exercise on brain aging remain largely unknown. Understanding the mechanisms of this protection may be the crucial first step in the development of much-needed therapies for neurological diseases associated with aging. One explanation for these benefits is that exercise improves mitochondrial health by both increasing mtDNA copy number and reducing mtDNA mutations in the brain. This increase in copy number ultimately increases energy availability and decreases the negative effects of inefficient metabolism, such as reactive oxygen species damage.

In PolG mice, exercise has been reported to improve many aspects of premature aging, including restoration of mtDNA copy number in skeletal muscle, myocardium, and liver tissue, which facilitates restoration of brain metabolism, normalization of protein levels, and preservation of oocyte quality [4,9–11]. Since the sole cause of premature aging in the PolG mouse model is an increase in mtDNA mutations, a decrease in mtDNA mutations would indicate an overall improvement and serve as a particularly reliable and relevant measure, or proxy, for mitochondrial health. Intermittent forced treadmill exercise has been reported to decrease mtDNA mutation load; however, subsequent studies were unable to replicate this finding [4,12]. A recent study using voluntary wheel running also found a decrease in mtDNA mutation load in skeletal muscle in the PolG mouse model, but failed to replicate the increased mtDNA copy number seen in prior studies [11]. To date, no studies of PolG mice have examined the effects of exercise on either mtDNA mutations or mtDNA copy number in the brain.

The primary aim of this study was to evaluate the impact of voluntary wheel running exercise on both mtDNA copy number and mutation measures, in the brain and, for comparison, in the liver of PolG mice. Because of the increases in corticosterone levels and anxiety behaviors observed in mice during forced exercise studies, and it's potential to interfere with any exercise benefits, including neurogenesis, we chose to implement a voluntary wheel running paradigm [13,14]. We hypothesized that voluntary exercise would ameliorate the mitochondrial abnormalities in the brain and liver, as has been suggested for other tissues.

In addition to assessing the effect of exercise on mtDNA in the PolG mouse, another goal of the study was to evaluate novel methods for measuring both mtDNA mutations and copy number to improve the efficiency and accuracy of measurements. Before sequencing to determine the abundance of mtDNA mutations, previous research has relied on costly and time-consuming methods to purify mtDNA. This step is taken to avoid including copies of mtDNA which have migrated into the nuclear genome (NUMTs)[15,16]. Concerns have been raised that NUMTs may affect mtDNA mutation data and that bioinformatics software is not able to accurately distinguish NUMTs from mtDNA [15]. To address these concerns, we employed next-generation sequencing (NGS) to measure mutations in both total genomic extract (unpurified) and purified mtDNA from the same tissue to evaluate the effect of purification on mtDNA mutation indices. We also used NGS to confirm the accuracy of our qPCR measurements of mtDNA copy number, as has been done previously [17,18]. Finally, we implemented a novel method of mtDNA purification that can be used to reduce sequencing costs and increase mtDNA coverage across a wide range of applications.

## 2. Methods

### 2.1 Animals

Male PolG mice (n = 34) were bred from heterozygous PolG mice. At two months of age, PolG mice were individually housed and randomly assigned to either an exercise group or sedentary group. Littermates were deliberately placed in opposite groups. Mice in the exercise group were housed in cages with 9-inch Mini-Mitter® running wheels, where wheel running was monitored continuously by VitalView® 4.1 software. Young male C57BL/6J mice (n = 10) between 2.5 and 6 months of age, and old C57BL/6J mice (n = 12) between 24 and 30 months of age, were used as wild-type controls, as they do not accumulate mtDNA mutations. Homozygous negative (HN), PolG mice, which do not possess the mutation, but which nevertheless inherit mtDNA mutations from their heterozygous mothers were included as an additional control (n = 5). The number of animals used for each experiment is indicated in the figure legends. Rooms were kept on a light-dark cycle with lights on at 07:00 and off at 19:00. All procedures were approved by the Institutional Animal Care and Use Committee at the University of Illinois Urbana-Champaign. Animals were housed in animal care facilities approved by the American Association for Accreditation of Laboratory Animal Care at the Beckman Institute for Advanced Science and Technology.

### 2.2 Tissue extraction

Mice were anesthetized by intraperitoneal injection of ketamine (100 mg/kg) and xylazine (3 mg/kg), followed by transcardial perfusion with chilled (4˚C) sucrose-based solution (in mM: 234 sucrose, 11 glucose, 26 $NaHCO_3$, 2.5 KCl, 1.25 $NaH_2PO_4$, 10 $MgCl_2$, 0.5 $CaCl_2$). Mouse brains, spleen, and liver were quickly extracted and flash frozen in isopentane on dry ice. Excess isopentane was removed before transferring the tissue to -80˚C until processing.

### 2.3 DNA preparation

Brain and liver tissue were thawed using warm 65˚C Ibc solution (in mM 200 sucrose, 10 Tris/ HEPES pH 7.4, 1 EGTA/TRIS pH 7.4, total pH 7.4) and homogenized in an iced, 7 ml Wheaton tissue grinder. The homogenate was centrifuged for 10 minutes at 17,000g and the supernatant discarded. For total genomic extract, the QIAamp Mini kit[TM] was used on half of the homogenate according to the manufacturer's instructions. The total genomic DNA extract was subjected to qPCR and next-generation sequencing to determine copy number.

The remaining homogenate was extracted using a QiaPrep Mini kit [TM] to purify mtDNA. For brain tissue, the manufacturer's directions were followed with the exception of adding 500 µl of 10% SDS (w/w) to the homogenate at 60˚C for 20 minutes before the addition of P2 buffer (Qiagen). This addition allowed the brain tissue to better solubilize with the Qiagen buffers. The extracted DNA was then cut with the restriction enzyme, NspI (NEB), segmenting the 16 kb mtDNA genome into 3 approximately equal fragments. Gel bands were extracted using the Zymoclean Large Fragment DNA Recovery (Zymo). This process further purifies the mtDNA. The purified mtDNA was submitted for sequencing to determine mtDNA mutation rate and load.

### 2.4 qPCR

To determine mtDNA copy number relative to genomic DNA, the total DNA extract was subjected to qPCR using the Biorad CFX Connect™ real-time PCR detection system with PowerUp™ SYBR® Green Master Mix (Applied Biosystems) in 10 µl reactions according to the manufacturer's directions. The qPCR reactions were performed in triplicate with primers

mB2MR1: 5' CAG TCT CAG TGG GGG TGA AT 3', mB2MF1: 5'ATG GGA AGC CGA ACA TAC TG 3', mMitoR1: 5' CCA GCT ATC ACC AAG CTC GT 3', mMitoF1: 5' CTA GAA ACC CCG AAA CCA AA 3' [19]. These primers were designed against mtDNA sequences not duplicated in the nuclear genome (NUMTs) and as such are likely to yield higher mtDNA copy number values than those seen in prior studies[19]. The qPCR temperature cycle was as follows: 1. 50˚C for 2:00, 2. 95˚C for 2:00, 3. 95˚C for 0:15, 4. 60˚C for 1:00, 5. Repeat 3–4 39x. Primer efficiency was calculated using the slope of the Cq vs. log(DNA concentration) line of four quintuplicate ten-fold dilutions. Efficiency = $-1+10^{(-1/slope)}$. For both primers, the $R^2$ value was >0.99. MtDNA copy number = $mB2MF_{efficiency}^{Average\ mB2MF\ Cq}$ / $mMito_{efficiency}^{Average\ mMito\ Cq}$. To verify our qPCR, we sequenced several full genomic extract samples and calculated the copy number from sequencing using the total mitochondrial genome coverage divided by the total nuclear genome coverage.

## 2.5 Mutation analysis

DNA samples were sequenced at the University of Illinois at Urbana-Champaign Illumina sequencing facility using high-depth, 250 nucleotide paired-end sequencing. The shotgun genomic libraries were prepared with the Hyper Library construction kit from Kapa Biosystems, in this case with no prior PCR amplification (Roche). The libraries were quantitated by qPCR and sequenced on one lane for 151 cycles from each end of the fragments on a HiSeq 4000 using a HiSeq 4000 sequencing kit version 1. Fastq files were generated and demultiplexed with the bcl2fastq v2.17.1.14 Conversion Software (Illumina). Adapters were trimmed using Trimmomatic and aligned using bwa with the GenCode mm10 mtDNA reference. Mutect2 and FilterMutectCalls (GATK 4.0) were used to call mutations in tumor-only mode using haploid settings, unlimited maximum events per region, max reads per alignment start at 5, and all other settings at default. Since mtDNA is circular, the alignment was performed a second time with the first 8000 basepairs of the reference cut and added to the end of the reference. Basepairs 1 through 500 and 15,800–16,299 were patched from the second alignment. Two ways of analyzing the mtDNA mutations were performed. **Mutation rate** is defined as the fraction of basepair positions in the mtDNA genome that had a mutation called (Number of basepair positions with a mutation called/16299bp). **Mutation load** is defined as the total number of mutations over the total number of basepairs sequenced (Number of mutations/ total number of bases sequenced). Germline and somatic mutations were also separated. Mutations that were shared between the brain and liver tissue were classified as germline [20]. Thus, the germline mutation rate is the samefor brain and liver (Fig 3A)

## 2.6 Statistics

Data were analyzed using SAS (version 9.4) Proc MIXED. P ≤ 0.05 was considered statistically significant. Data were considered normally distributed if residual distribution showed skewness between -1 and 1 and kurtosis between -2 and 2. Body mass and spleen mass were analyzed by one-way ANOVA with 5 groups (homozygous negative, young wild type, old wild type, PolG sedentary and PolG exercised). Within PolG mice, body mass and spleen mass were also analyzed using analysis of covariance with age as a covariate and exercised/sedentary group as the factor. Average total revolutions per day were calculated for each individual for each month of the experiment and analyzed using repeated-measures ANOVA with month as the repeated measures factor to establish whether running changed as the animals aged. mtDNA measures (copy number, mutation rate and load) were compared between exercised and sedentary PolG mice using 2-way repeated-measures ANOVA with tissue (liver or brain) entered as the within-subjects factor and group (exercised versus sedentary) as the between-

subjects factor. mtDNA copy number was also compared across all subjects using group (young wild type, old wild type, or PolG collapsed across sedentary and exercised) as the between-subjects factor. Within PolG mice, mtDNA measures in liver and brain were separately correlated with spleen mass (health indicator) and age. In addition, a linear model was constructed predicting spleen mass with covariates age, copy number or mutation rate/load, to determine any impact of mtDNA measures on spleen mass after correcting for the age-spleen mass relation. Finally, within exercised PolG mice, mtDNA measures were correlated with total distance run on running wheels, and distance run in the last month of running. ANOVAs were followed by Least Significant Difference (LSD) tests, to evaluate post-hoc pair-wise differences between means. Consistency in estimates of copy number from qPCR and sequencing were assessed using Pearson's correlation.

## 3 Results

### 3.1 Body mass

The one-way ANOVA yielded a significant effect of group ($F_{4,50}$ = 22.7, P<0.0001; Fig 1A). PolG mice (exercised or sedentary) were approximately 30% lighter than wild-type (young or old) mice or homozygous negative controls (all P<0.0001). No other differences were detected. Within the PolG mice, older animals displayed significantly lower body mass than younger animals as indicated by a significant effect of the age covariate ($F_{1,30}$ = 5.6, P = 0.02; Fig 1A), and after correcting for age, exercised mice had approximately 10% lower body mass than sedentary mice ($F_{1,30}$ = 4.1, P = 0.05). This physical effect of wheel running on body mass is evidence that the mice were running sufficiently to generate a physiological impact.

### 3.2 Spleen mass

The one-way ANOVA revealed a significant effect of group ($F_{4,51}$ = 4.4, P = 0.004; Fig 1B). PolG mice (both exercised and sedentary) and old wild-type mice displayed an enlarged spleen compared to homozygous negative mice and young wild-type (all P<0.04). No other differences were detected. Within the PolG mice, spleen mass was positively correlated with age ($F_{1,29}$ = 14.0, P< 0.0008), and was not affected by running.

### 3.3 Wheel running data

As young adults, PolG mice ran approximately 4 km/day which is typical of standard inbred strains [21], and similar to what was recently reported for PolG mice [11]. Wheel running decreased significantly as the animals aged, also typical for mice [22], and the PolG model [11]. This decrease was indicated by a significant effect of age in the repeated measures ANOVA ($F_{9,104}$ = 61.0, P <0.0001, Fig 1C).

### 3.4 mtDNA copy number

Using copy number estimated from qPCR, and considering PolG mice only, no difference in copy number was detected between sedentary and exercised animals. Moreover, no correlation was detected between cumulative distance traveled (or distance traveled the last month) and mtDNA copy number. Hence PolG sedentary and runners were combined. In an analysis considering young wild-type, old wild-type, and PolG mice, collapsed across sedentary and exercised, copy number was significantly greater in liver than brain ($F_{1,42}$ = 33.9, P<0.0002; Fig 2A), and there was an effect of group ($F_{2,45}$ = 4.2, P = 0.02; Fig 2A), but no interaction between group and tissue type. Post-hoc tests indicated that PolG mice had greater mtDNA copy number than old wild-type (P = 0.01). Taken together, these results demonstrate that PolG mice do

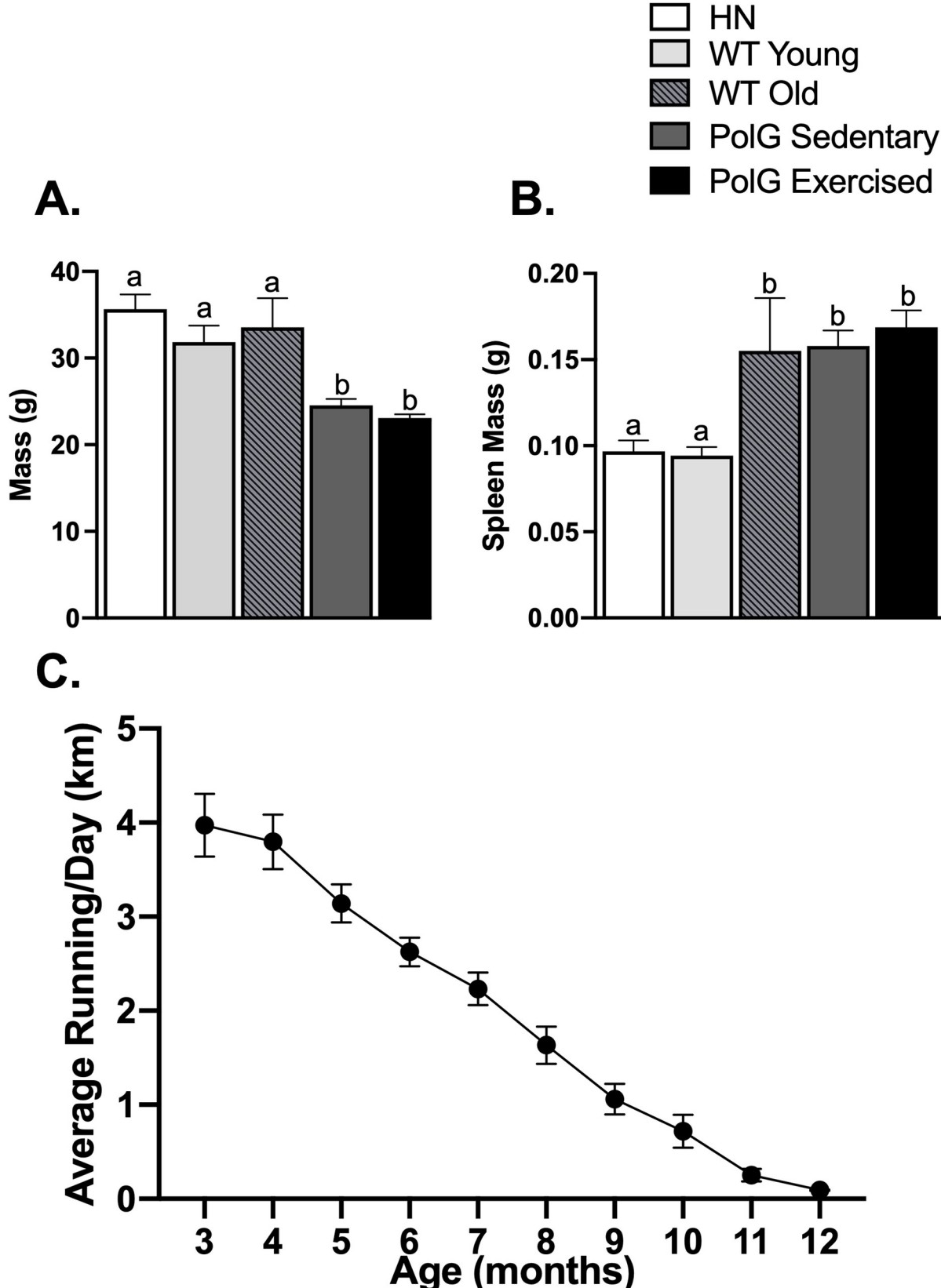

**Fig 1. Physical measures in exercised and sedentary PolG animals.** Mean A. body mass and B. spleen mass of PolG homozygous negative (HN), young wild type (WT), old wild type (WT), and PolG sedentary and exercised animals (n = 3, 8, 10, 18, 16 per group, respectively for body mass; n = 5, 8, 10, 17, 16 per group, respectively for spleen mass). C. Mean distance run (km/day) at each month of age in PolG animals. Month 3 represents data accumulated between 60 and 90 days of age for each animal, etc. Month 3 (n = 15; one animal was not recorded by accident), month 4–8 (n = 16), month 9 (n = 15), month 10 (n = 11), month 11 (n = 6), month 12 (n = 2). Cohorts of mice were euthanized between ages 9 and 12 months, which accounts for the reduction in sample size at the different months. Different letters (a vs b) denote P < 0.05 significance; all error bars show ± SEM.

not show premature aging in the mtDNA copy number phenotype in the brain or liver. Copy number from sequencing and copy number from qPCR were strongly correlated (r = 0.94, P<0.0003; Fig 2B), suggesting the sequencing method works as well as the qPCR standard.

### 3.5 Mutation load and rate in PolG mice

The total mutation load and rate were approximately 19% and 65% greater in liver samples than brain, respectively, collapsed across exercised and sedentary (Fig 3A and 3B). These increases were indicated by a significant main effect of tissue type for mtDNA mutation load ($F_{1,13}$ = 10.1, P = 0.007) and rate ($F_{1,13}$ = 166.2, P<0.0001). No effect of exercise or interaction between exercise and tissue type was detected. Moreover, within PolG exercised mice, no correlation was observed between cumulative distance traveled (or distance traveled in the last month) and mutation load or rate. In addition, the age covariate was not significant.

Germline mutations accounted for approximately 55% of brain mutation rate and 33% of liver mutation rate. Germline mutations were more dominant in the mutation load metric, accounting for 85% of the total load in brain and 71% in liver. For the somatic mutation rate and load, a two-way ANOVA showed a significant effect of tissue (rate, $F_{1,13}$ = 164.3, P< 0.0001; load $F_{1,13}$ = 16.3, P<0.001), but no effect of exercise, or interaction between exercise and tissue. For reference, one young wild-type (3 months), one old wild-type (25 months), and one PolG homozygous negative brain were sequenced (Fig 4). While wild-type animals showed a significantly lower mutation rate and load compared to the germline and total mutations of PolG animals, the homozygous negative animal showed a total mtDNA mutation load and rate very similar to the germline mutation load and rate in the PolG mutator animals (Fig 4).

### 3.6 Spleen mass correlations

Within PolG mice (exercised and sedentary combined), spleen mass was significantly correlated with liver copy number (*r* = 0.42, P = 0.02; Table 1), but not brain copy number. Copy number in liver and brain samples of PolG mice were not significantly correlated with age. Liver mutation rate and load were significantly correlated with spleen mass (rate, *r* = 0.65, P = 0.007, load, *r* = 0.50, P = 0.05; Table 1). Brain mtDNA mutation rate was also significantly correlated with spleen mass (*r* = 0.58, P = 0.02; Table 1), but brain mtDNA mutation load was not (*r* = 0.44, P = 0.09; Table 1).

### 3.8 mtDNA sequencing data and NUMTs

The purified mtDNA samples have 16x more mtDNA mapped reads than the total genomic extract. (P = <0.0001; Fig 5). There is no significant difference between the purified and unpurified samples in mutation rate or mutation load. The mutation rates and mutation loads from the same sample (r = 0.9936, 0.9614; Fig 5) are highly and significantly correlated between the purified and unpurified samples. Tissue samples were taken from sedentary, PolG brains.

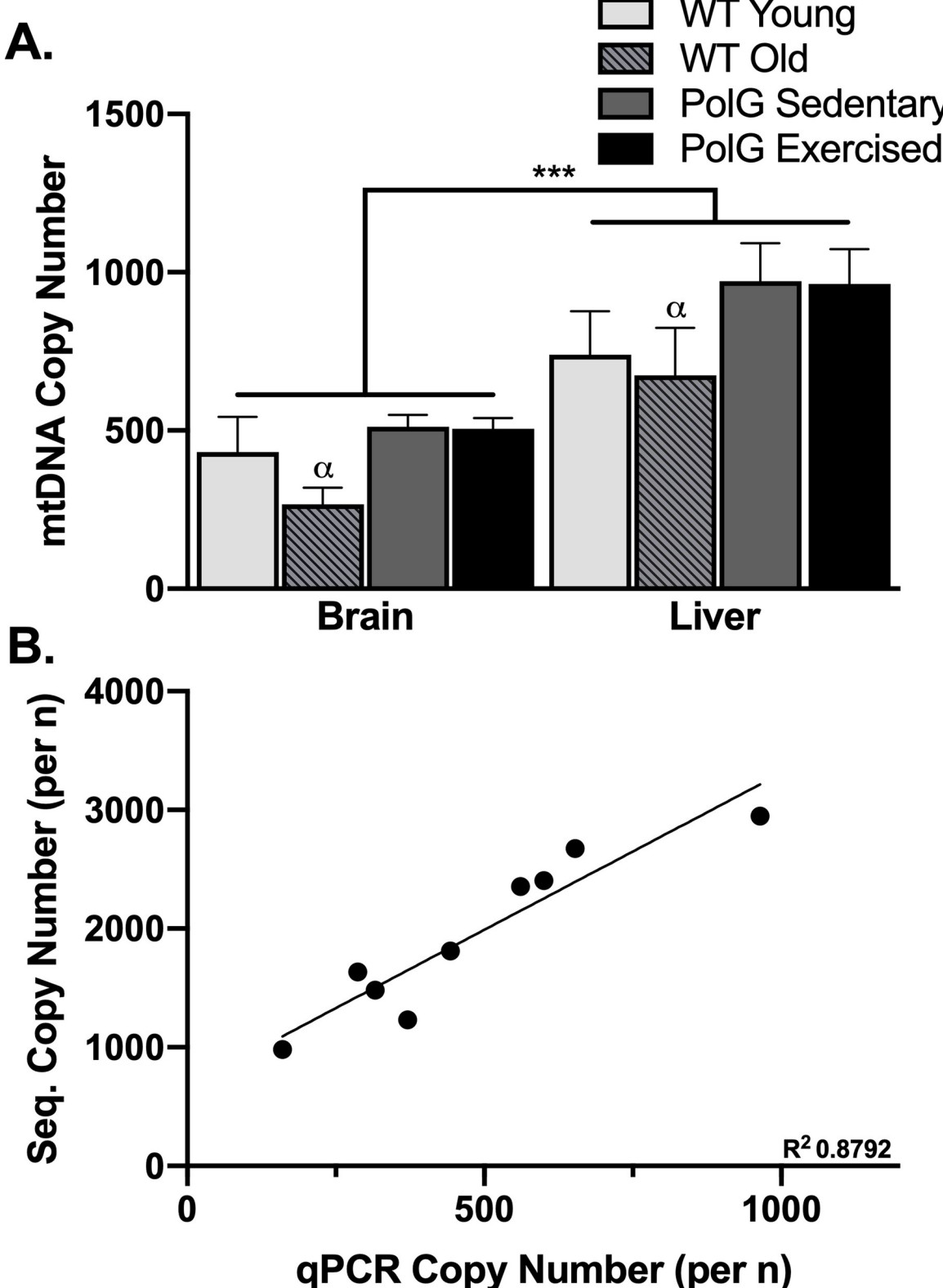

**Fig 2. Copy number in PolG animals.** A. Mean copy number in samples from brain and liver from young wild-type (WT) (n = 6 for brain and n = 7 for liver), old wild-type (WT) (n = 6 for both liver and brain), exercised PolG (n = 18 for both liver and brain), and sedentary PolG mice (n = 16 for liver and brain). α P < 0.01 for comparison of old wild- type (WT) to PolG within brain and liver, and ***P < 0.001 for comparison between brain and liver; all error bars show ± SEM. B. Correlation of copy number calculated from sequencing vs. copy number calculated from qPCR.

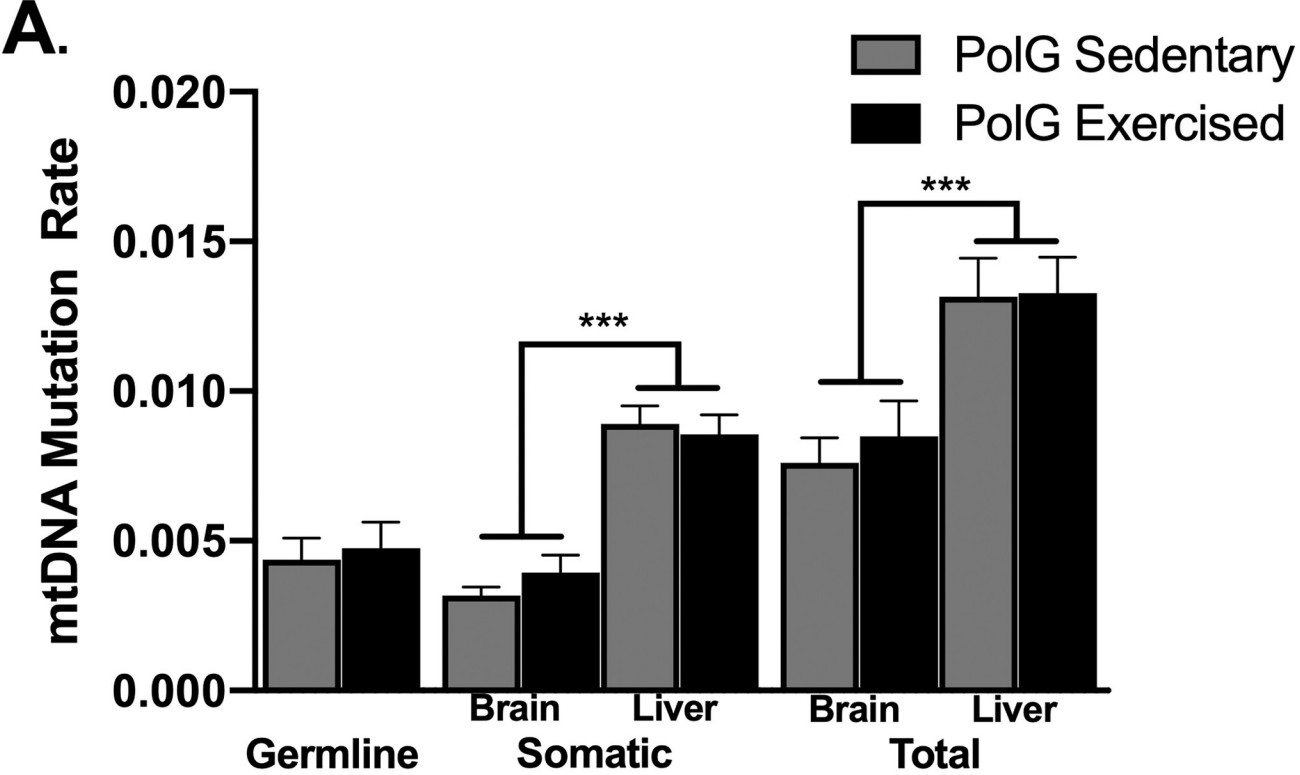

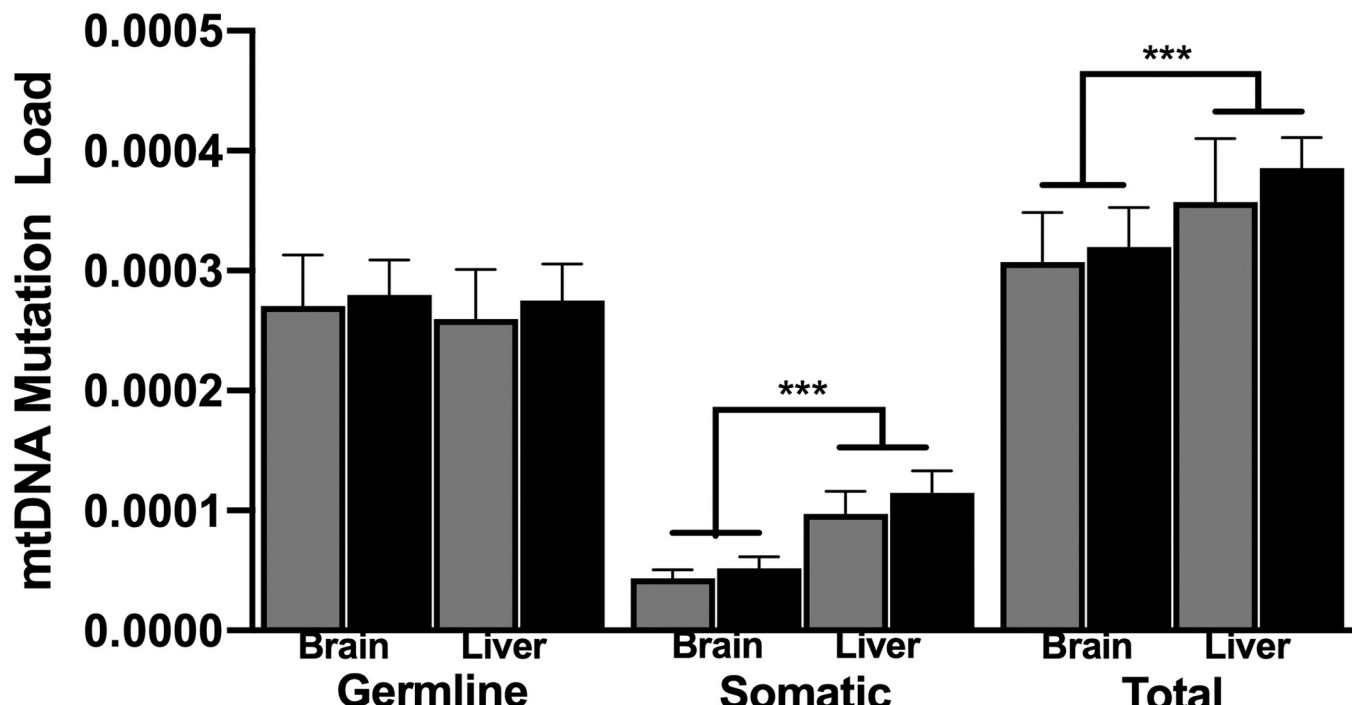

**Fig 3. mtDNA mutation analysis in PolG mice.** A,B. Mean mtDNA mutation rate and load in samples from brain and liver from exercised and sedentary PolG mice (germline and somatic: PolG Sedentary (n = 8), PolG Exercised (n = 7); Total: Brain PolG Sedentary (n = 9), else (n = 8). ***P < 0.001; all error bars show ± SEM.

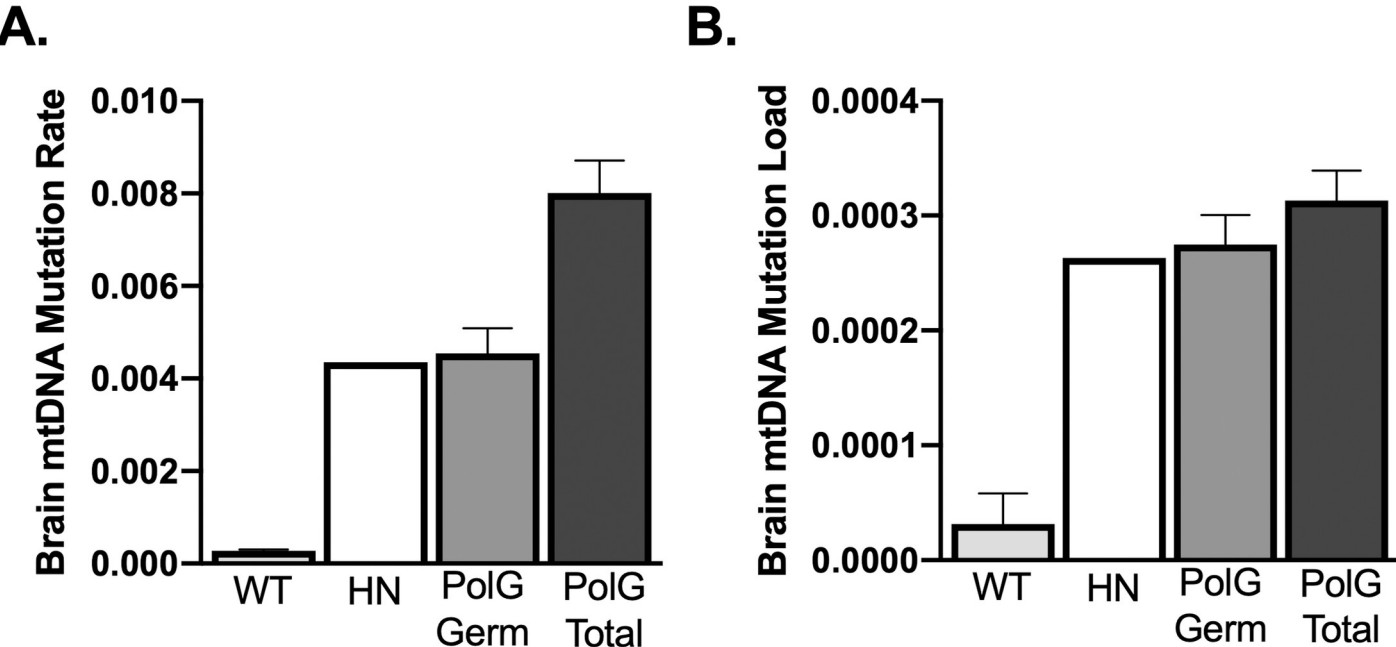

**Fig 4. mtDNA mutation analysis in HN and WT mice.** A, B. Relative comparison between wild type (WT) (n = 2 composed of one young (3 months) and one old (25 months), homozygous negative (HN) (n = 1 (12 months)), and PolG (germ n = 15, total n = 17) brain mtDNA mutation rate and load, collapsed across exercised and sedentary groups.

## 4 Discussion

The current study finds no evidence that voluntary wheel running has an impact on mtDNA copy number, mutation rate, or mutation load in the brain or liver in the PolG mouse model of premature aging. This lack of effect occurred in spite of significant decreases in body mass with exercise. These findings suggest that the effects of exercise in delaying cognitive aging may be unrelated to changes in mitochondrial mutations or copy number in the brain and may be mediated by different mechanisms influenced by exercise, such as protein regulation

**Table 1. Age and spleen mass correlations with mtDNA measures.** A. Pearson's correlations between spleen mass, age and mtDNA measures. B. Pearson's correlations between age and mtDNA measures.

| Spleen Mass vs.: | N | R | P value |
|---|---|---|---|
| Age | 33 | 0.5525 | **0.0009**\*\*\* |
| Liver Copy Number | 33 | 0.4173 | **0.016**\* |
| Brain Copy Number | 33 | 0.2316 | 0.19 |
| Liver Mutation Rate | 16 | 0.6478 | **0.0067**\*\* |
| Liver Mutation Load | 16 | 0.4966 | **0.05**\*\* |
| Brain Mutation Rate | 16 | 0.5805 | **0.018**\* |
| Brain Mutation Load | 16 | 0.4416 | 0.087 |
| Age vs.: | N | R | P value |
| Liver Copy Number | 33 | 0.2245 | 0.20 |
| Brain Copy Number | 33 | 0.1394 | 0.43 |
| Liver Mutation Rate | 16 | 0.0802 | 0.93 |
| Liver Mutation Load | 16 | 0.2566 | 0.34 |
| Brain Mutation Rate | 16 | 0.1909 | 0.46 |
| Brain Mutation Load | 16 | 0.3953 | 0.12 |

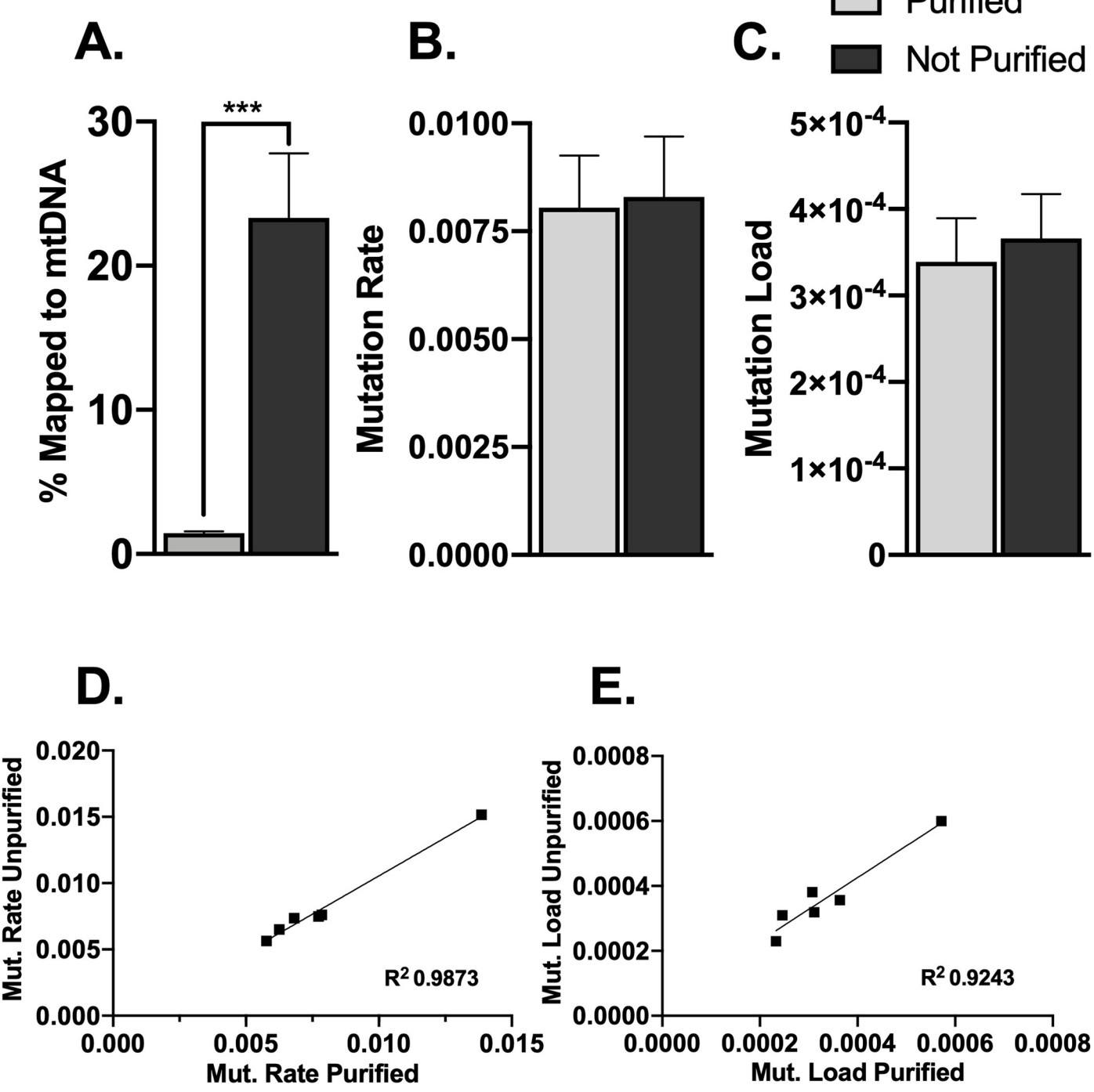

**Fig 5. mtDNA purification and NUMTs.** A. Percent of reads mapped to the mtDNA reference in purified and unpurified total genomic extract. B. Mutation rate in purified and unpurified samples. C. Mutation load in purified and unpurified samples. D,E. Correlations between purified and unpurified mutation rate and mutation load.(N = 6) *P < 0.05, **P < 0.01, and ***P < 0.001; all error bars show ± SEM.

[11] or adult hippocampal neurogenesis [23]. To the best of our knowledge, this is the first time mtDNA mutation and copy number measures have been reported for brain tissue in the PolG model. We expected exercise would ameliorate the mutant mitochondrial phenotypes in the brain as it has been reported to do in other tissues such as liver and muscle [4,11].

However, our results for the brain and liver are consistent with other PolG studies that have found no effect of forced treadmill exercise on mtDNA mutation load in muscle or oocytes [9,12]. Studies that reported reduced mtDNA mutations in muscle with exercise were either unable to be replicated or only a small portion of the mitochondrial genome was explored [4,12]. Differences in tissue, regions of sequencing, exercise method and age at examination may explain these seemingly contrasting results. Although in our study, exercise did not appear to alleviate aging phenotypes such as an increase in spleen mass, an increase in mtDNA mutations, or a decrease in mtDNA copy number in the PolG mouse, it does not conflict with data showing anti-aging benefits from exercise, as these benefits likely come from mechanisms other than changes in mtDNA mutations.

It is possible that stress, in the form of corticosterone levels, could negate some or all of the expected benefits of exercise in both the brain and periphery. Due to an increase in corticosterone levels and anxiety behaviors previously observed in mice during forced exercise studies, which may interfere with any exercise benefits, we chose voluntary running using a running wheel [13]. Although the voluntary running paradigm necessitated housing animals individually, which has been shown to increase stress in some studies while decreasing it in others, wheel running ensured that the exercise itself would not induce stress and allows for both a direct measurement of running distance (for a single animal) and more opportunity for exercise (without competition) [24,25]. Additionally, many different strains of non-mutant laboratory mice display large increases in hippocampal neurogenesis from running when singly housed [21], so the effect of social isolation on exercise-induced brain adaptations in mice is not sufficient to outweigh certain benefits. It is unlikely that our result is due to lack of exercise as the young adult mice began running at an average of 4 km per day which is typical of standard inbred strains [21,26]. In addition, in our exercised PolG animals, a lowered body mass was observed after correcting for age, demonstrating that the animals exercised at a sufficient level for physiological impact. Moreover, a recent study of PolG mice showed positive effects of voluntary wheel running on multiple exercise-related phenotypes and those PolG mice ran at levels very similar to the mice in this study [11]. It is also unlikely that null findings were due to a lack of robust methods, as we employed next-generation sequencing on purified mtDNA samples to achieve a high level (over 500x) of coverage to find mtDNA mutations, which were later further separated into germline and somatic mutations. We also compared our qPCR measurements to next-generation sequencing results on the same samples, verifying their accuracy.

## 4.1 Copy number effects

Other papers examining the effects of exercise on mtDNA copy number in the brain of wild-type mice have reported mixed results, possibly due to the different methodologies employed and use of different sets of primers. Increased mtDNA copy numbers, strongest in the brainstem and cortex, were seen in young mice in response to forced treadmill running [27]. An increase in mtDNA copy number in brain was also seen in aged mice in response to high intensity (supra-lactate) forced running [28]. Intriguingly, the same authors have reported decreases in copy number in the livers of young mice [29]. In humans, both control subjects and patients with chronic obstructive pulmonary disease show temporary reductions in mtDNA copy number which correlates with tissue reactive oxygen species production in response to supra-lactate threshold exercise, with sub-lactate exercise reducing the strength of this association. These changes are nevertheless concurrent with increases in PGC-1$\alpha$ and elevations of anti-oxidant enzymes [30]. Consistent with the current findings, others have also reported no change in brain mtDNA copy number in response to treadmill running [31]. In

all cases, changes are not dramatic and near the threshold of significance, thus our findings are not that dissimilar.

These data also add to the current understanding of PolG model in the context of aging more generally. Previous studies found that mtDNA copy number decreases with age [5,32]. Our results showed slightly decreased mtDNA copy number with age in wild-type mice, though it was not statistically significant. Previous studies reported decreased mtDNA copy number in PolG mice as an indicator of premature aging when compared to homozygous negative littermates in heart, liver, and skeletal muscle [4,11]. While we found no difference in mtDNA copy number between PolG mice and young wild-type mice, but a decrease in mtDNA copy number in old wild-type mice, it is important to note that our wild-type mice were not from the PolG strain, as in prior studies [4,12]. Therefore, in the aspect of mtDNA copy number, PolG mice do not appear to show premature aging when compared to wild-type animals from outside of the PolG strain. This finding indicates that a reduction in mtDNA copy number in liver or brain is not necessary to display an aged phenotype.

## 4.2 Mutation rate and load

Although we extensively analyzed mtDNA mutations by measuring both mutation rate, which reflects de novo mutations, and mutation load, which reflects the spread of mutations, we did not find that exercise impacted either of these mtDNA mutation measures in PolG mice. We also further analyzed mtDNA mutations by separating somatic and germline mutations. The inclusion of both mutation rate and mutation load, and the separation of germline and somatic mtDNA mutations, adds to the understanding of how mtDNA mutations might affect health. Mutation rate was more strongly correlated with spleen mass than load. Spleen mass has been shown to be a good indicator of aging and its associated pathologies, where larger spleens are associated with infections, vascular alterations, autoimmune disorders, hematologic malignancies and metabolic syndromes [33]. Therefore, mutation rate may be a better indicator of health in aging than mutation load. This difference may explain why homozygous negative animals have a normal phenotype with a high mtDNA mutation load and low mutation rate. This result also suggests that there may be selection in the way that mtDNA mutations spread in an organism. One explanation for these data is that silent or neutral mutations spread throughout the body and across generations more easily than harmful mutations, as harmful mutations would cause more immediate apoptosis preventing their spread. Hence, PolG animals with their deficiency in preventing de novo somatic mutations are harmed more than homozygous negative animals who merely inherited the germline mutations that survived selection but did not produce any de novo mutations of their own.

As seen in previous studies, the liver showed both greater mtDNA copy number and greater mutations in both measures than did the brain [2,19]. The increase in mtDNA mutations in liver tissue may be due to a greater cell turnover rate in liver than brain, since more replication of mtDNA generates more opportunity for mutation [34,35]. This suppostion is supported by findings that show a further increase in the duodenum, a tissue with an even greater turnover rate than liver [3,34,36]

## 4.3 Novel methodology and utility of mtDNA purification

Many primers designed for measuring mtDNA copy number contain mtDNA sequences that have migrated into the genome (NUMTs). The primers used in our study map to regions of the mitochondrial genome not duplicated in the nuclear genome. Our values per given tissue are commensurate with values reported in the original paper which introduced these primers [19]. Our estimations of mtDNA copy number via qPCR were also confirmed by comparison

to NGS. The strong correlation between copy number found from qPCR and sequencing demonstrates that our qPCR was performed accurately and can reliably support conclusions using relative mtDNA copy number. The discrepancy between the absolute data from sequencing and qPCR may be due to NUMTs being mapped to the mitochondrial DNA reference.

Previous mtDNA studies have emphasized the importance of purifying mtDNA to avoid NUMT contamination [15,16]. There has been a concern that similar sequences in the mtDNA genome and the nuclear genome will affect the mtDNA mutation data. However, our results showed no significant difference in mutation rate or load from purified mtDNA and unpurified genomic extract. This is an important contribution because total genomic extractions are less expensive and more efficiently generated. If the total genomic extract is submitted for sequencing, these data can also be used to find mtDNA copy number forgoing the need for qPCR. This allows mtDNA copy number and mtDNA mutation data to be collected in one step. Though mtDNA purification does not appear to affect mtDNA mutation data, our novel method is still helpful for achieving more mtDNA coverage per given use of a sequencing lane and may be more cost-effective for investigators sequencing large numbers of subjects.

## 5 Conclusions

We find that exercise has no impact on mtDNA mutation rate, mutation load, or copy number in liver and brain tissue. Our result implies that anti-aging benefits from exercise may come from a mechanism other than decreases in mtDNA mutations. The two metrics of mtDNA mutations employed here: mutation rate and mutation load, appear to have different health-related correlations in PolG mice. Namely, the mutation rate may better reflect the aging status of the PolG animal than mutation load. Though previous concerns have been raised about NUMTs affecting mtDNA mutation data, our results indicate that purification of mtDNA does not impact mutation data from next-generation sequencing and as such, future researches may be spared these laborious steps.

## Supporting information

**S1 Dataset.**
(XLSX)

## Acknowledgments

We thank the DNA Services Lab (University of Illinois at Urbana-Champaign) for sequencing the DNA samples. We thank HPCBio (University of Illinois at Urbana-Champaign) for assistance in processing the sequencing output.

## Author Contributions

**Conceptualization:** Kevin A. Stebbings.

**Data curation:** Kendra D. Maclaine, Kevin A. Stebbings.

**Formal analysis:** Kendra D. Maclaine, Kevin A. Stebbings, Justin S. Rhodes.

**Funding acquisition:** Kendra D. Maclaine, Kevin A. Stebbings, Daniel A. Llano.

**Investigation:** Kendra D. Maclaine, Kevin A. Stebbings.

**Methodology:** Kendra D. Maclaine, Kevin A. Stebbings.

**Project administration:** Kevin A. Stebbings.

**Resources:** Kendra D. Maclaine, Kevin A. Stebbings, Daniel A. Llano, Justin S. Rhodes.

**Supervision:** Kevin A. Stebbings, Daniel A. Llano, Justin S. Rhodes.

**Visualization:** Kendra D. Maclaine.

**Writing – original draft:** Kendra D. Maclaine, Kevin A. Stebbings.

**Writing – review & editing:** Kendra D. Maclaine, Kevin A. Stebbings, Daniel A. Llano, Justin S. Rhodes.

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
