## [Decision Letter · Decision Letter 0]

20 Dec 2019

PONE-D-19-33772

Voluntary wheel running has no impact on brain and liver mitochondrial DNA copy number or mutation measures in the PolG mouse model of aging

PLOS ONE

Dear Dr. Rhodes:

Thank you for submitting your manuscript to PLOS ONE. After careful consideration, we feel that it has merit but does not fully meet PLOS ONE’s publication criteria as it currently stands. Therefore, we invite you to submit a revised version of the manuscript that addresses the points raised during the review process.

We would appreciate receiving your revised manuscript by Feb 03 2020 11:59PM. To enhance the reproducibility of your results, we recommend that if applicable you deposit your laboratory protocols in protocols.io, where a protocol can be assigned its own identifier (DOI) such that it can be cited independently in the future. For instructions see: http://journals.plos.org/plosone/s/submission-guidelines#loc-laboratory-protocols

We look forward to receiving your revised manuscript.

Kind regards,

Jianhua Zhang

Academic Editor

PLOS ONE

Journal Requirements:

Reviewers' comments:

Reviewer's Responses to Questions

**Comments to the Author**

1. Is the manuscript technically sound, and do the data support the conclusions?

Reviewer #1: Yes

2. Has the statistical analysis been performed appropriately and rigorously? 

Reviewer #1: Yes

3. Have the authors made all data underlying the findings in their manuscript fully available?

Reviewer #1: Yes

4. Is the manuscript presented in an intelligible fashion and written in standard English?

Reviewer #1: Yes

5. Review Comments to the Author

Reviewer #1: This is a well-designed study to assess whether exercise can counteract the effects of increased mtDNA mutations induced by PolG mutation. The design and methods are thoroughly described, and this reviewer is particularly pleased with the amount of detail included in the statistical analyses. Though the majority of the data presented are 'negative', the robust experimental design and inclusion of multiple controls speaks to the reliability of the data presented. Further, these data answer the question the authors posed, and is of value to the research community.

I have no major concerns with this manuscript but recommend addressing these minor points in a revision:

- did the authors consider assessing corticosterone levels in the young vs old animals? As the authors are investigating effects of exercise in the brain, they implemented voluntary wheel-running to minimize stress, which necessitated animals being housed alone. Mice are social mammals and there is evidence of increased corticosterone in single-housed rodents. This stress response may also be age-dependent and worth measuring if plasma/serum samples are available.

- I strongly recommend a thorough copy-edit of the manuscript by the senior author to ensure consistency in acronyms, hyphens, and general writing style. The English is technically correct, but could be edited for style.

6. PLOS authors have the option to publish the peer review history of their article (what does this mean?). If published, this will include your full peer review and any attached files.

Reviewer #1: No

---

## [Author Response · Author response to Decision Letter 0]

29 Jan 2020

Thank you for organizing the review process. Below the comments from the REVIEWER ARE IN BOLD and our response is in plain text.

REVIEWER #1: THIS IS A WELL-DESIGNED STUDY TO ASSESS WHETHER EXERCISE CAN COUNTERACT THE EFFECTS OF INCREASED MTDNA MUTATIONS INDUCED BY POLG MUTATION. THE DESIGN AND METHODS ARE THOROUGHLY DESCRIBED, AND THIS REVIEWER IS PARTICULARLY PLEASED WITH THE AMOUNT OF DETAIL INCLUDED IN THE STATISTICAL ANALYSES. THOUGH THE MAJORITY OF THE DATA PRESENTED ARE 'NEGATIVE', THE ROBUST EXPERIMENTAL DESIGN AND INCLUSION OF MULTIPLE CONTROLS SPEAKS TO THE RELIABILITY OF THE DATA PRESENTED. FURTHER, THESE DATA ANSWER THE QUESTION THE AUTHORS POSED, AND IS OF VALUE TO THE RESEARCH COMMUNITY.

I HAVE NO MAJOR CONCERNS WITH THIS MANUSCRIPT BUT RECOMMEND ADDRESSING THESE MINOR POINTS IN A REVISION:

- …DID THE AUTHORS CONSIDER ASSESSING CORTICOSTERONE LEVELS IN THE YOUNG VS OLD ANIMALS? AS THE AUTHORS ARE INVESTIGATING EFFECTS OF EXERCISE IN THE BRAIN, THEY IMPLEMENTED VOLUNTARY WHEEL-RUNNING TO MINIMIZE STRESS, WHICH NECESSITATED ANIMALS BEING HOUSED ALONE. MICE ARE SOCIAL MAMMALS AND THERE IS EVIDENCE OF INCREASED CORTICOSTERONE IN SINGLE-HOUSED RODENTS. THIS STRESS RESPONSE MAY ALSO BE AGE-DEPENDENT AND WORTH MEASURING IF PLASMA/SERUM SAMPLES ARE AVAILABLE.

We very much appreciate the reviewer’s affirmation of our attempts to produce a study with careful design, statistical power, and methodology despite the ultimately negative results. We also, of course, hope these results are useful to scientists interested in exercise and/or mitochondria.

Unfortunately plasma samples are not available to test for corticosterone levels, but probably would have been a good idea in retrospect. The effects of stress are a particularly interesting parallel pathway in the exercise literature. As the reviewer points out, many exercise studies, including this one, probably do not sufficiently address the potential effects of stress responses, both in relation to cortisol/corticosterone and in other pathways. Indeed, the effects of social isolation are likely a confounder in many studies, not just exercise studies. Thus, we added a paragraph to the discussion (second paragraph) to address the concerns about single housing and stress.

The measuring of corticosterone levels during or across exercise interventions would likely provide a useful covariate to explore. For instance, it may be paradoxically possible that the positive effects of exercise seen in the forced treadmill running vs our free-wheel running may, in fact, be due to the severe stress and not to the exercise itself.

For consistency, sedentary animals were also housed individually, so any stress caused by social isolation was also experienced by the sedentary animals. Overall, we felt that the stress caused by forced exercise would have caused an unequal and severe amount of stress for the exercised group. Anecdotally, there is also evidence that male mice specifically experience less stress when housed individually (perhaps due to fighting) and all of the mice in the current study were male. Our rodent facility also only allows direct littermates of male mice to be co-housed due to fighting concerns which can result in forced euthanasia by veterinarian order (statistically risky). Since we split littermates deliberately into the 2 groups to attempt to control for litter effects, co-housing littermates to prevent social isolation would not be simultaneously possible. 

- I STRONGLY RECOMMEND A THOROUGH COPY-EDIT OF THE MANUSCRIPT BY THE SENIOR AUTHOR TO ENSURE CONSISTENCY IN ACRONYMS, HYPHENS, AND GENERAL WRITING STYLE. THE ENGLISH IS TECHNICALLY CORRECT, BUT COULD BE EDITED FOR STYLE.

We have done our best to edit the manuscript for consistency and have, hopefully, improved the general style. 

The question of hyphenation, particularly of “wild-type” vs “wild type” caused us some reflection, as the first is appropriate to an adjective (whether the corresponding noun is present or implied), the second to a noun. We have used the un-hyphenated form only in the case where “wild type” is the name of a category in statistical analysis, that is, a group name.

---

## [Decision Letter · Decision Letter 1]

14 Feb 2020

Voluntary wheel running has no impact on brain and liver mitochondrial DNA copy number or mutation measures in the PolG mouse model of aging

PONE-D-19-33772R1

Dear Dr. Rhodes:

We are pleased to inform you that your manuscript has been judged scientifically suitable for publication and will be formally accepted for publication once it complies with all outstanding technical requirements.

With kind regards,

Jianhua Zhang

Academic Editor

PLOS ONE

Additional Editor Comments (optional):

Reviewers' comments:

Reviewer's Responses to Questions

**Comments to the Author**

1. If the authors have adequately addressed your comments raised in a previous round of review and you feel that this manuscript is now acceptable for publication, you may indicate that here to bypass the “Comments to the Author” section, enter your conflict of interest statement in the “Confidential to Editor” section, and submit your "Accept" recommendation.

Reviewer #2: All comments have been addressed

2. Is the manuscript technically sound, and do the data support the conclusions?

Reviewer #2: Yes

3. Has the statistical analysis been performed appropriately and rigorously? 

Reviewer #2: Yes

4. Have the authors made all data underlying the findings in their manuscript fully available?

Reviewer #2: Yes

5. Is the manuscript presented in an intelligible fashion and written in standard English?

Reviewer #2: Yes

6. Review Comments to the Author

Reviewer #2: I have no comments. The manuscript is well written and the experimental design is excellent. This study is a worthwhile contribution to the literature.

7. PLOS authors have the option to publish the peer review history of their article (what does this mean?). If published, this will include your full peer review and any attached files.

Reviewer #2: No

---

## [Editor Report · Acceptance letter]

19 Feb 2020

PONE-D-19-33772R1 

Voluntary wheel running has no impact on brain and liver mitochondrial DNA copy number or mutation measures in the PolG mouse model of aging 

Dear Dr. Rhodes:

I am pleased to inform you that your manuscript has been deemed suitable for publication in PLOS ONE. Congratulations! Your manuscript is now with our production department. 

With kind regards,

on behalf of

Dr Jianhua Zhang 

Academic Editor

PLOS ONE